# The Mid-Upper Arm Circumference as a Substitute for Body Mass Index in the Assessment of Nutritional Status among Pregnant Women: A Cross-Sectional Study

**DOI:** 10.3390/medicina59061001

**Published:** 2023-05-23

**Authors:** Yasir Salih, Saeed M. Omar, Nadiah AlHabardi, Ishag Adam

**Affiliations:** 1Faculty of Medicine, University of Khartoum, P.O. Box 102, Khartoum 11111, Sudan; dryasirsalih@yahoo.ie; 2Faculty of Medicine, Gadarif University, Gadarif 32211, Sudan; 3Department of Obstetrics and Gynecology, Unaizah College of Medicine and Medical Sciences, Qassim University, Unaizah 51911, Saudi Arabia

**Keywords:** pregnancy, body mass index, mid-upper arm circumference, undernutrition, obesity

## Abstract

To compare mid-upper arm circumference (MUAC) with body mass index (BMI) and propose MUAC cut-off points corresponding to BMIs of <18.5 kg/m^2^ (underweight) and ≥30.0 kg/m^2^ (obesity) for pregnant Sudanese women, a cross-sectional community-based study was conducted in the city of New Halfa, Sudan. Healthy pregnant women were recruited. Body parameters such as height, weight, and MUAC were measured. The MUAC (cm) cut-off values for underweight and obesity were determined using receiver operating characteristic (ROC) curve analysis. Of 688 pregnant women, 437 were in early pregnancy (<20 weeks of gestation) and 251 were in late pregnancy (≥20 weeks of gestation). There was a significant positive correlation between BMI and MUAC among women in both early pregnancy (*r* = 0.734) and late pregnancy (*r* = 0.703). The cut-off points of MUAC for detecting underweight and obesity were found to be 24.0 cm and 29.0 cm, respectively, for women in early pregnancy, with good predictive values. For women in late pregnancy, the cut-off points for detecting underweight and obesity were 23.0 cm and 28.0 cm, respectively. We concluded that for Sudanese pregnant women, the MUAC cut-off points identified in the study for diagnosing underweight and obesity are both sensitive and specific.

## 1. Introduction

The prevalence of malnutrition, which presents as both underweight and obesity, is a problem worldwide, and it is more common in resource-poor countries [1,2]. The assessment of nutritional status during pregnancy is of vital importance for ensuring optimal health outcomes for both the fetus and the mother. Being underweight or obese is associated with several unfavorable pregnancy outcomes such as preeclampsia, gestational diabetes, low birth weight, small-for-gestational-age newborns, and preterm labor [3]. Identifying the nutritional status of pregnant women is a critical aspect of prenatal care. The most commonly used anthropometric method for assessing nutritional status is the measurement of the body mass index (BMI). In this method, the height and weight of the pregnant woman are measured, and the BMI is calculated using the formula: BMI = weight (in kilograms)/(height (in meters))^2^ [4]. Based on the BMI, the subject is classified as underweight (BMI < 18.5 kg/m^2^); within the normal range (BMI = 18.5 to 24.9 kg/m^2^); overweight (BMI > 24.9 to 29.9 kg/m^2^); or obese (BMI ≥ 30 kg/m^2^) [5]. This technique requires some training and specific calibrated medical weighing scales and stadiometers. Several attempts have been made to find alternatives by using other anthropometric measurements [6]. Circumference measurements such as those of the arm, abdomen, and thigh and skinfold thickness measurements are among the recommended methods [7]. Several studies have shown a significant correlation between the values of BMI and measurements of the mid-upper arm circumference (MUAC) of adults in different populations [6,8,9]. Previous studies have shown varying degrees of correlation between BMI and MUAC during pregnancy [10,11]. Moreover, cut-off points of MUAC have been reported to detect underweight and obesity during pregnancy [10,11]. Due to differences in the anthropometric measurement standards within different populations, there is an urgent need to identify local cut-off points to be used during prenatal care.

Both low weight and obesity are common health problems among pregnant Sudanese women [12,13], and they are associated with several adverse effects such as gestational diabetes, macrosomia, and increased rates of cesarean delivery and delivery of a low-birthweight newborn [14,15]. We recently showed that MUAC can be used to assess underweight and obesity among non-pregnant Sudanese adults [6]. This study aimed to compare MUAC with BMI and propose MUAC cut-off points corresponding to BMIs of <18.5 kg/m^2^ (underweight) and ≥30.0 kg/m^2^ (obesity) for pregnant Sudanese women.

## 2. Materials and Methods

A cross-sectional study was conducted in the city of New Halfa, eastern Sudan, from January to February 2021. We adopted a multistage sampling technique adhering to the Strengthening the Reporting of Observational Studies in Epidemiology (STROBE) Statement standard checklists [16]. The details of the study were described previously [6]. In summary, the city of New Halfa is divided into seven sectors, of which we selected four using simple random sampling, and within the total sample size of the study [17], the four sections were distributed according to the size allocation of the sector. When a selected household did not contain a pregnant woman or did not agree to participate in the study, then the next household was chosen.

The inclusion criteria were a multistage sampling method for selection, healthy pregnant women with singleton pregnancies, residents of New Halfa. and consent for participation. The exclusion criteria were pregnant women with any chronic diseases such as diabetes, thyroid diseases, and heart failure, pregnant women who were critically ill or had a severe acute illness, women with any pregnancy complications, and women who refused to participate. Tokens were not provided during recruitment or participation in the study. Two female medical officers (trained for measurements over 5 days) performed the measurements in full privacy.

Based on the selection criteria, the women signed informed consent, and their age, parity, and the date of their last menstrual period were recorded. The fundal height was determined via clinical examination. This was followed by weight measurements after removing shoes and any heavy clothing or objects using a standard medical weighing scale. Height was measured using a stadiometer with bare feet and the subject positioned to assure that the back of the heels, buttocks, and shoulder blades touched the back plate with the head in the Frankfurt horizontal plane. The MUAC was measured using a non-stretchable MUAC measuring tape. The mid-point between the left acromion and the olecranon was identified to measure the MUAC. The arthrometric measurements were taken twice, and the average was taken. When there was a discrepancy between the two measurements, a third one was taken. The instruments used to perform these measurements were calibrated daily. BMI was calculated using the standard formula: weight in kg/(height in m)^2^ [18]. The BMI cut-off points of <18.5 kg/m^2^ and ≥30.0 kg/m^2^ were used to identify adult underweight and obese women, respectively.

A sample of 688 pregnant women (in early or late gestational age) were calculated to show the significant minimum difference in the correlation (r = 0.11) between BMI and measured MUAC. This sample (688 pregnant women) had an 80% power and a difference of 5% at α = 0.05 [19]. These pregnant women were selected from all sectors in New Halfa so as to ensure state-wide representativeness, and then the sample size (688) was divided according to the proportion of the sector population. 

Data were collected and statistical analysis was performed using the Statistical Package for the Social Sciences (SPSS, Version 22.0). The normality distribution was tested using Shapiro–Wilk tests, and the data were not normally distributed. The Mann–Whitney U was performed to assess the differences in variables between women in early and late pregnancy. Descriptive statistics were obtained for the demographic variables and all measurements (parity, BMI, and MUAC). Scatter plots with fitted linear regression lines were computed to evaluate the association between MUAC and BMI among the women in early and late pregnancy, respectively. The sensitivity and the specificity were computed. Youden’s index (YI) was calculated as YI = sensitivity + specificity − 1. The MUAC cut-off with the highest YI value represented the optimal statistically derived cut-off [20]. The area under the receiver operating characteristic curve (AUROCC with its 95% confidence interval, CI) was obtained for underweight and obesity in early and late gestational age, respectively. *p* < 0.05 was considered statistically significant.

## 3. Results

In our study, 688 pregnant women agreed to participate. The medians (IQR) for age, parity, gestational age, MUAC, and BMI were 26 (15–37) years, 3 (0–6), 17 (8–26) weeks, 25 (19–31) cm, and 23.81 (17.71–29.91) kg/m^2^, respectively, as shown in Table 1. Out of the 688 participating women, 90 (12.95%) were underweight (BMI < 18.5 kg/m^2^) and 103 (14.85%) were obese (BMI ≥ 30.0 kg/m^2^). Of these 688 pregnant women, 437 and 251 were in early (<20 weeks of gestation) and late pregnancy (≥20 weeks of gestation), respectively. There was no significant difference between the profiles (age, parity, MUAC, and BMI) of the two groups, as seen in Table 1.

There was a significant positive correlation between the BMI and the MUAC values among the women in early pregnancy (*r* = 0.734, *p* < 0.001) and women in late pregnancy (*r* = 0.703, *p* < 0.001), as shown in Figure 1.

Among women in early pregnancy, the cut-off point of MUAC for detecting underweight was 24.0 cm (YI = 0.57; sensitivity = 71.0%, specificity = 86.0%), with a good predictive value (AUROCC = 0.83; 95.0% CI = (0.79–0.88), as shown in Table 2 and Figure 2. On the other hand, for women in late pregnancy, the cut-off point of MUAC for detecting obesity was 29.0 cm (YI = 0.67; sensitivity = 80.0%, specificity = 87.0%), with a good predictive value (AUROCC = 0.90; 95.0% CI = (0.86–0.94), as shown in Table 2 and Figure 2.

In women in late pregnancy, the cut-off point of MUAC for detecting underweight was 23.0 cm (YI = 0.57, sensitivity = 84.0%, specificity = 72.0%), with a good predictive value (AUROCC = 0.81; 95.0% CI = (0.70–0.91), as shown in Table 2 and Figure 2. Meanwhile, for women in early pregnancy, the cut-off point of MUAC for detecting obesity was 28.0 cm (YI = 0.61; sensitivity = 76.0%, specificity = 86.0%), with a good predictive value (AUROCC = 0.89; 95.0% CI = (0.83–0.95), as shown in Table 2 and Figure 2.

## 4. Discussion

The main finding of the current study is the positive correlation between MUAC and BMI in pregnant women in both early (*r* = 0.734) and late gestational age (*r* = 0.703). Previously, Fakier et al. reported similar findings in terms of a positive correlation between MUAC and BMI in early pregnancy as well as late pregnancy in South Africa (*r* = 0.93, *r* = 0.92) [10]. In a large study including 2912 pregnant women which was conducted in London and Dublin, Cooley et al. reported a positive correlation between MUAC and BMI (*r* = 0.836) [21]. Recently, based on the findings of a study including 1165 pregnant women attending a prenatal session in Brazil, Miele et al. reported a strong correlation between MUAC and BMI in early pregnancy (*r* = 0.872), mid-pregnancy (*r* = 0.870), and late pregnancy (*r* = 0.831) [11]. Moreover, strong correlations between MUAC and BMI were previously reported among non-pregnant women in Iran (r = 0.91) [22] as well as non-pregnant women in India (*r* = 0.86) [23].

In our study, the cut-off point for detecting underweight using MUAC was 24.0 cm for women in early gestational age and 23.0 cm for those in late gestational age. Previously, Fakier et al. proposed a cut-off point of 22.8 cm to identify pregnant women under 30 weeks of gestational age as underweight in South Africa [10]. However, a higher cut-off (25.75 cm) was reported for identifying underweight pregnant women with a gestational age of 19–21 weeks in Brazil [11]. The same cut-off point of 25.75 cm was also found to detect underweight during late pregnancy in Brazil [11]. Moreover, cut-off values of 24.0 cm and 23.0 cm were reported to determine non-pregnant women as underweight in Iran [22] and India [23], respectively. Interestingly, we recently observed that the MUAC cut-off point of 23.0 cm was associated with low-birth-weight delivery in Sudan [15]. In neighboring Ethiopia, the same MUAC cut-off point of 23.0 cm was associated with low-birth-weight delivery [24].

Our results showed that the cut-off point for detecting obesity was 29.0 cm in early gestational age and 28.0 cm in late gestational age. The MUAC cut-off for obesity among pregnant women in South Africa was 30.5 cm, which was rounded to 31.0 cm for practical applications [10]. Moreover, higher cut-offs of 30.15 cm for early gestational age and 30.6 cm for late gestational age were reported to detect obesity among pregnant women in Brazil [11].

It is worth mentioning that caution must be practiced when comparing our results with those of later studies. Firstly, the difference in the prevalence of underweight and obesity among pregnant women in different settings could explain the difference in the results, e.g., the fact that only 1.8% of women in South Africa were underweight [10]. This difference in the anthropometric measurement standards could be explained by sociodemographic, nutritional, and genetic factors. Secondly, we assessed the MUAC cut-off points using Youden’s index, which we also used recently for assessing MUAC cut-offs in non-pregnant adults [6] but was not used in the later studies which assessed MUAC cut-offs [10,11].

The MUAC measure is a useful alternative to BMI. It has been reported that the cost of the equipment used to measure weight and height in order to compute BMI is roughly 1500 times more expensive than the cost of the MUAC measuring tape [23]. Likewise, the time needed to perform the MUAC measurement is less than that required for measuring weight and height [24].

Although the findings of this study can be considered as a starting point for establishing local reference values to detect underweight and obesity among pregnant women using MUAC in Sudan, it still has its limitations. The study was conducted in one city in the eastern part of Sudan, which may not be a fair representation of the entire population, considering the vast area of the country. Furthermore, the number of participants in each category (obesity and underweight) may not be large enough to establish cut-off points that are acceptably projectable to the entire country.

## 5. Conclusions

This study confirms the strong positive correlation between BMI and MUAC, and therefore, the MUAC can be used reliably as an indicator and alternative to BMI measurements. For Sudanese pregnant women, we recommend using the MUAC cut-off points of 24.0 cm in early pregnancy and 23.0 cm in late pregnancy, respectively, to detect underweight. Furthermore, the MUAC cut-offs of 29.0 cm in early pregnancy and 28.0 cm in late pregnancy are both reasonably sensitive and specific for diagnosing obesity. Further larger cohort studies are recommended.

## Figures and Tables

**Figure 1 medicina-59-01001-f001:**
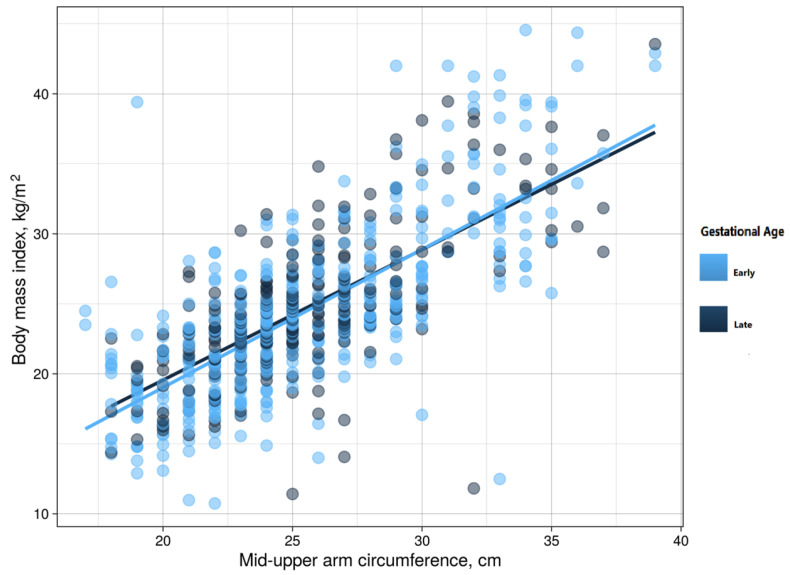
Correlations between mid-upper arm circumference and body mass index among pregnant women in eastern Sudan, 2021.

**Figure 2 medicina-59-01001-f002:**
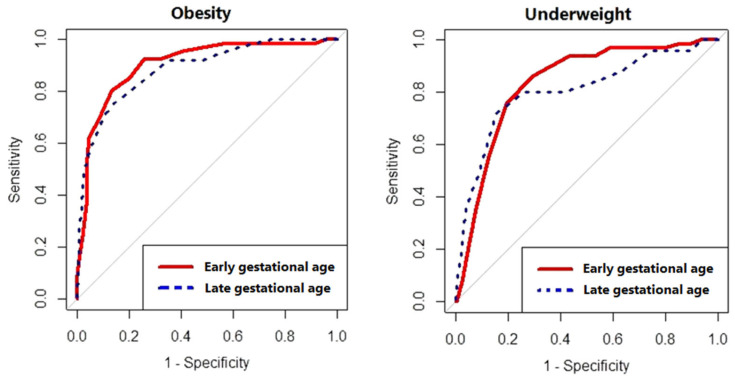
Performance of mid-upper arm circumference in detecting underweight and obesity among pregnant women in eastern Sudan, 2021.

**Table 1 medicina-59-01001-t001:** Comparison of medians (interquartile range) in the profiles of women in early and pregnancy in eastern Sudan, 2021.

Variables	Total (*n* = 688)	Women in Early Pregnancy (*n* = 437)	Women in Late Pregnancy (*n* = 251)	*p*-Value
Age, years	26.0 (15.0–37.0)	26.0 (20.0–31.0)	26.0(21.0–30.0)	0.574
Parity	3 (0–6)	3 (1–4)	3 (1–4)	0.875
Body mass index, kg/m^2^	23.8 (17.7–29.9)	23.6 (20.5–27.1)	24.0(21.5–26.9)	0.157
Mid-upper arm circumference, cm	25.0 (19.0–31.0)	24.0 (22.0–28.0)	25.0(23.0–27.0)	0.393
Gestational age, weeks	17.0 (8.0–26.0)	13.0 (10.0–17.0)	23.0(21.0–30.0)	<0.001

**Table 2 medicina-59-01001-t002:** Performance of mid-upper arm circumference in the diagnosis of underweight and obesity among pregnant women in eastern Sudan, 2021.

Variables		Early Pregnancy	Late Pregnancy
Underweight	The area under the receiver operating characteristic curve	0.83	0.81
	Cut-off	24.0 cm	23.0 cm
	Sensitivity	71.0	85.0
	Specificity	86.0	72.0
	Positive predictive value	97.0	96.0
	Negative predictive value	34.0	34.0
	Youden’s index	0.57	0.57
Obesity	The area under the receiver operating characteristic curve	0.90	0.88
	Cut-off	29.0 cm	28.0 cm
	Sensitivity	80.0	76.0
	Specificity	87.0	86.0
	Positive predictive value	51.0	47.0
	Negative predictive value	96.0	95.0
	Youden’s index	0.67	0.61

## Data Availability

The datasets used and/or analyzed during the current study are available from the corresponding author upon reasonable request.

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
