# Peer review of "The Mid-Upper Arm Circumference as a Substitute for Body Mass Index in the Assessment of Nutritional Status among Pregnant Women: A Cross-Sectional Study"

_medicina, 2023, doi:10.3390/medicina59061001_

Round 1

Reviewer 1 Report

Dear authors,

Please address the following comments:

1. Line 59, please include what these adverse effects are.

2. Include the Research design and sample size calculation.

3. Please clarify if tokens were provided during recruitment or in participating in the study. 

Good luck!

Author Response

We would like to thank the editor and reviewers for their valuable comments on the MS. We feel that their comments and the response have improved the MS 1 dramatically

Reviewer # 1

Comment

. Line 59, please include what these adverse effects are.

Response

Yes, agreed and we have included them as suggested” effects such as gestational diabetes, macrosomia, increased rate of cesarean delivery and delivering a low birthweight newborn: Please see lines 59- 60

Comment

2. Include the Research design and sample size calculation.

Response

Yes, we agreed and we included them as suggested. Please see line 67 and lines 97 -103

Comment

3. Please clarify if tokens were provided during recruitment or in participating in the study.

Response

Tokens were provided, this has been inserted. Please see line 81.

Reviewer 2 Report

This a cross sectional study of mid-upper arm circumference of pregnant women from randomly selected households in New Haifa Sudan.  Measurements were correlated with BMI and established cut-offs for underweight status and obesity. 

1.       In the materials and methods, add who did the measurements and how they were trained.

2.       Was this study performed contemporaneously with the study from reference 6?  If so, that should be mentioned

3.       What distinguishes this study from other studies on MUAC, is it only the location and is it important to research the numbers from each country.

4.       How is this study generalizable to a larger group?

Author Response

We would like to thank the editor and reviewers for their valuable comments on the MS. We feel that their comments and the response have improved the MS 1 dramatically

Reviewer # 2

Comment

1. In the materials and methods, add who did the measurements and how they were trained.

Response

2. Two female medical officers (trained for measurement for 5 days) performed the measurements in full privacy.  Please see line 80-81

Comment

3. Was this study performed contemporaneously with the study from reference 6?  If so, that should be mentioned

Response

Yes, and this point has been inserted as suggested. Please see line 70

Comment

4. What distinguishes this study from other studies on MUAC, is it only the location and is it important to research the numbers from each country.

Response

5. There are several differences between our study and the other studies. This has been mentioned.Please see lines 191-199.

Comment

6. How is this study generalizable to a larger group?

Response

7. Yes, it can be generalized (with caution). This point has been inserted. Please line see 208- 210.

Regards
